# Impact of a Short-Term Physical Activity Program on Emotion Regulation and Eating Behaviors Among Technical University Students

**DOI:** 10.3390/healthcare13202621

**Published:** 2025-10-18

**Authors:** Ofelia Popescu, Valentina Stefanica, Halil İbrahim Ceylan, Marko Joksimović, Nicoleta Leonte, Daniel Rosu

**Affiliations:** 1Department of Physical Education and Sports-Kinetotherapy, Faculty of Medical Engineering, National University of Science and Technology Politehnica Bucharest, 060042 Bucharest, Romania; ofelia.popescu@upb.ro; 2Department of Physical Education and Sport, Faculty of Sciences, Physical Education and Informatics, National University of Science and Technology Politehnica Bucharest, Pitesti University Center, 110040 Pitesti, Romania; daniel.rosu@upb.ro; 3Physical Education of Sports Teaching Department, Faculty of Sports Sciences, Ataturk University, 25240 Erzurum, Türkiye; 4Faculty of Sport and Physical Education, University of Montenegro, 81000 Podgorica, Montenegro; marko.j@ucg.ac.me

**Keywords:** self-regulation, dietary patterns, exercise intervention, university student well-being, higher education, psychological resilience, sedentary behavior

## Abstract

**Background:** Emotion regulation (ER) difficulties are closely linked to maladaptive coping strategies, including impulsive and emotional eating, which undermine health and well-being in young adults. Technical university students are particularly vulnerable due to factors such as a high academic workload, sedentary behavior, and performance-related stress. This study evaluated the effects of a four-week structured physical activity intervention on ER and eating behaviors among engineering students. **Methods:** Seventy first- and second-year computer science and engineering students (40 males and 30 females, aged 19–25 years) from Politehnica University of Bucharest participated in the study. The intervention included three weekly supervised training sessions and a daily step count requirement (≥6000 steps), verified via weekly smartphone submissions. Pre- and post-intervention assessments employed the Difficulties in Emotion Regulation Scale (DERS-36) and the Adult Eating Behavior Questionnaire (AEBQ-35). Data were analyzed using Kolmogorov–Smirnov tests, Wilcoxon signed-rank tests, and paired-sample *t*-tests. **Results:** Significant improvements were observed in five ER domains—non-acceptance of emotional responses, goal-directed behavior, impulse control, access to regulation strategies, and emotional clarity (all *p* < 0.01). No change occurred in emotional awareness (*p* > 0.05). Eating behaviors (restrained, emotional, and external eating) showed no significant differences pre- and post-intervention (all *p* > 0.05). **Conclusions:** A short-term, structured physical activity program enhanced emotion regulation capacities but did not alter eating behaviors in the short run. These findings highlight the feasibility of embedding low-cost, exercise-based modules into higher education to strengthen students’ psychological resilience. Longer and multimodal interventions may be required to produce measurable changes in eating behaviors.

## 1. Introduction

Emotion regulation (ER) has gained substantial recognition in psychology and the health sciences and is defined as the processes through which individuals monitor, evaluate, and modify their emotional responses to achieve adaptive functioning [1,2,3]. Contemporary perspectives emphasize that ER is a motivated, goal-directed process shaped by regulatory goals and individual differences, with difficulties in ER linked to maladaptive behaviors such as impulsivity and disordered eating [4,5].

Eating behaviors represent another critical determinant of health in young adults, with three constructs widely studied: Restrained Eating (cognitive control of food intake), Emotional Eating (food intake driven by emotions rather than hunger), and External Eating (eating in response to external food cues) [6,7,8]. Difficulties in ER have been strongly associated with maladaptive eating patterns. For instance, Miller and Racine (2020) [9] found that lack of emotional clarity predicted binge eating among students, while Yang et al. (2023) [10] and Howells et al. (2023) [11] confirmed positive associations between ER difficulties and emotional eating. Gerges et al. (2022) [12] further demonstrated that ER difficulties mediated the link between maladaptive schemas and disordered eating in young adults. These findings suggest that ER deficits and problematic eating patterns often co-occur, undermining psychological and physical well-being.

Students in technical fields, such as computer science and engineering, may be particularly vulnerable due to high academic workload, prolonged sedentary behavior, and performance-related pressures. University students often experience high academic pressure and lifestyle disruptions, which increase their vulnerability to mental health problems such as anxiety, depression, and stress [13]. Danowitz and Beddoes (2022) [14] reported higher rates of mental health problems among first- and second-year engineering students, especially women, while Campos-Uscanga et al. (2023) [15] observed high levels of sedentary behavior and irregular eating patterns in this population.

Research on Romanian university students reveals concerning patterns of physical inactivity, stress, and maladaptive coping behaviors [16,17]. Academic overload is evident through high levels of academic burnout, which exceeds work-related burnout among employed students and contributes to academic maladjustment mediated by test anxiety [18]. These findings collectively highlight the multifaceted challenges that Romanian university students face regarding their physical health, mental well-being, and academic performance.

In response, interventions aimed at fostering self-regulation and resilience have gained attention. Research examining eating behaviors among university students reveals complex relationships between psychological, social, and behavioral factors that may explain the failures of interventions. Cross-sectional studies demonstrate that entrenched psychological traits and social factors significantly influence eating behaviors [19]. Mindfulness-based approaches reduce emotional eating and improve coping [20], while structured physical activity programs, including multimodal formats like Cross-Fit, have demonstrated significant benefits for both physical and psychological health [21,22,23]. The integration of such interventions in university contexts has shown improvements in mood, stress reduction, and social connectedness [24,25].

Taken together, these findings indicate that emotion regulation deficits and maladaptive eating behaviors frequently co-occur and may reinforce one another, with ER difficulties increasing susceptibility to emotional and external eating [26,27]. Investigating these constructs jointly allows for a more comprehensive understanding of students’ self-regulatory capacities and their behavioral manifestations under academic stress.

The primary aim of this study was to evaluate the impact and measurable outcomes of a four-week structured physical activity intervention on emotion regulation and eating behaviors among first- and second-year computer science and engineering students at Politehnica University of Bucharest. The program, which combined supervised training sessions with daily step monitoring, was designed to strengthen self-regulation capacities and promote healthier coping strategies in response to academic stressors [28].

Our research hypotheses:

1. Students participating in the intervention will demonstrate significant improvements in multiple dimensions of emotion regulation between pre- and post-intervention assessments.

2. The intervention is expected to reduce maladaptive eating patterns (restrained, emotional, and external eating), although short-term effects may be limited due to the stability of dietary habits.

3. Students exposed to the intervention will show improvements in adaptive self-regulatory processes (e.g., goal-directed behaviors, impulse control, access to regulation strategies), which can be interpreted as indicators of more effective coping with academic demands.

## 2. Materials and Methods

### 2.1. Study Design

This study employed a quantitative, pre–post intervention design without a control group. Although the research included an intervention and repeated measures, it does not constitute a randomized experimental design in the strict methodological sense, as it was conducted between February and May 2024. The investigation was structured into sequential phases to ensure methodological rigor, with an emphasis on measuring changes in emotional regulation and eating behaviors following a structured intervention program.

A convenience sampling strategy was used, as participants were recruited from existing Physical Education classes and through online announcements on the university’s internal platforms. All students enrolled in the Faculty of Automation and Computers were invited to participate voluntarily. This approach was chosen because the target population shared comparable academic and lifestyle routines, thereby minimizing variability due to external factors [29].

### 2.2. Procedures

The research unfolded in several consecutive stages:

1. Recruitment and Enrollment (February 2024). Students from the Faculty of Automation and Computers at Politehnica University of Bucharest were invited to participate voluntarily. Recruitment took place during Physical Education classes and through university announcements. All participants signed written informed consent, and no incentives were offered.

2. Baseline Assessment (26 February–10 March 2024). Participants completed two validated self-report instruments: the Difficulties in Emotion Regulation Scale (DERS-36) and the Adult Eating Behavior Questionnaire (AEBQ-35). These tools were selected for their strong psychometric properties and relevance to the constructs under study [30,31].

3. Initial Data Processing (February–March 2024). Baseline data were centralized and screened to ensure quality and completeness.

4. Participant Grouping (1 March 2024). Following baseline assessment, students were not randomly assigned to groups; instead, all participants who met the inclusion criteria were included in a single intervention group. Allocation was therefore based solely on eligibility (age range, enrollment in the faculty, compliance with step count monitoring, and informed consent).

5. Training Intervention (March–April 2024). The experimental phase consisted of a structured physical training program specifically designed for the study. The regimen consisted of a combination of aerobic, strength, and functional exercises, performed three times per week over four consecutive weeks. The sessions emphasized progressive overload and included both individual and group-based components.

6. Final Assessment (1 May 2024). At the end of the intervention, participants completed the same validated questionnaires used at baseline, ensuring comparability of measures.

7. Final Data Processing (3 May 2024). Collected data were compiled, anonymized, and prepared for statistical analysis.

8. Data Analysis (3–30 May 2024). Comparative analyses were conducted to evaluate changes in outcomes before and after the intervention.

### 2.3. Participant Selection, Grouping, and Socio-Demographic Characteristics

Participants were eligible for inclusion if they were aged between 19 and 25 years and were enrolled as first- or second-year students in the Faculty of Automation and Computers. These criteria were chosen to ensure a homogeneous sample of young adults with comparable academic and lifestyle demands, thereby facilitating a valid analysis of emotional regulation and eating behaviors [32,33]. Exclusion criteria included prior participation in structured physical or psychological programs aimed at improving emotional regulation, failure to self-report daily step counts (with a minimum threshold of 6000 steps/day verified through weekly photographic submissions and short feedback questionnaires), or refusal to provide written informed consent.

Out of 123 students who initially expressed interest, 100 attended the recruitment phase. After screening, 30 were excluded: 15 for exceeding the age range, 14 for failing to submit step count reports, and 1 for declining consent. The final sample consisted of 70 students (40 males and 30 females) who completed the baseline assessment. All 70 participants also completed the final post-intervention assessment, as no participants dropped out during the study.

#### Additional Contextual Variables

In addition to the standardized instruments (DERS-36 and AEBQ-35), a short contextual questionnaire was administered at baseline to collect descriptive information on psychological distress, academic workload, dietary habits, cultural norms, and societal expectations. These variables were not measured using validated international scales. Still, they were operationalized through single-item, self-report questions, explicitly developed for this study, to better describe the sample context (Table 1).

Responses were coded into categorical variables and presented in Table 1 to provide descriptive information on the sample. Since these indicators were intended to describe the initial socio-demographic and psychosocial profile, they were not reassessed after the intervention, and therefore, no changes over time are reported. Most students resided in urban areas and lived in dormitories or rented accommodations. Many reported moderate to high levels of psychological distress and perceived their academic workload as demanding. Dietary patterns were heterogeneous, with a substantial proportion of participants reporting irregular eating habits or consuming fast food. Cultural and societal influences, particularly pressure to maintain body image, were also prevalent. These factors provided critical context for interpreting the intervention’s outcomes.

Before participating in the study, each participant provided written consent, ensuring compliance with the Declaration of Helsinki and adherence to ethical standards. Students were thoroughly informed about the study’s objectives, methods, and any potential risks and benefits associated with it. Consent was collected before the research began, confirming their voluntary participation. The study received ethical clearance from the Ethics Committee of the Doctoral School of Physical Education and Sport Science at Politehnica University of Bucharest (Approval ID: 12/28.01.2024), ensuring the safeguarding of research integrity and participant rights.

### 2.4. Research Instruments

This study employed two validated self-report questionnaires to assess participants’ difficulties with emotion regulation and eating behavior patterns. Both measures have been extensively applied in previous student-focused research. They are particularly suitable for assessing subjective mental health and lifestyle-related variables within the framework of exercise-based interventions. To date, no officially validated Romanian versions of these instruments exist; therefore, the original English versions were used in this study. Before the baseline assessment, the questionnaires were administered in their original English version. Since English is not the native language of the participants, a pre-test was conducted with a group of 10 students from the same faculty to verify the comprehensibility of the items. Students were asked to report any difficulties in understanding specific words or formulations. Minor clarifications were provided verbally during the pre-test session (e.g., synonyms or equivalent Romanian expressions for less familiar terms), but no structural changes were made to the instruments.

Feedback from the pre-test confirmed that participants could reliably understand and respond to the English items, given their advanced English proficiency as computer science and engineering students. Consequently, the full versions of DERS-36 and AEBQ-35 in English were used for the baseline and post-intervention assessments.

#### 2.4.1. Difficulties in Emotion Regulation Scale (DERS-36)

The DERS-36 is a 36-item self-report instrument developed to assess multiple facets of emotional dysregulation [34]. It measures an individual’s ability to recognize, understand, and effectively manage emotional experiences across six domains: nonacceptance of emotional responses, difficulties engaging in goal-directed behavior, impulse control difficulties, lack of emotional awareness, limited access to regulation strategies, and lack of emotional clarity. Items are rated on a Likert scale, with higher scores indicating greater difficulties in emotion regulation. The scale is widely recognized for its strong reliability and construct validity, providing comprehensive insights into the regulation challenges commonly associated with psychological distress [30,35].

#### 2.4.2. Adult Eating Behavior Questionnaire (AEBQ-35)

The AEBQ-35 is a 35-item self-report instrument that evaluates appetite-related traits and eating styles across two higher-order dimensions: Food Approach (e.g., food responsiveness, hunger, emotional overeating, enjoyment of food) and Food Avoidance (e.g., satiety responsiveness, food fussiness, emotional undereating, slowness in eating) [31]. Items are rated on a 5-point Likert scale ranging from “Strongly Disagree” to “Strongly Agree.” Higher scores on Food Approach subscales indicate greater sensitivity to food stimuli and a stronger appetite-driven behavior. In comparison, higher scores on Food Avoidance subscales suggest stronger tendencies toward restrictive or selective eating. The AEBQ-35 has been widely applied in research exploring appetite regulation, food-related decision-making, and their connections to psychological well-being [31,36,37].

### 2.5. Experimental Intervention Program

The four-week intervention combined supervised training sessions with daily step monitoring, specifically designed to influence both emotion regulation (ER) and eating behaviors. The structured exercise sessions aimed to enhance emotional resilience by strengthening self-efficacy, impulse control, and stress management, while the step-monitoring component fostered accountability and healthier routines that could indirectly affect maladaptive eating patterns. Given the short duration, stronger effects were anticipated for ER than for eating behaviors; nevertheless, assessing both provided insight into the broader self-regulatory impact of physical activity.

Participants attended four weekly sessions (≈45 min each) that included aerobic, strength, and functional mobility exercises, with intensity progressively adjusted to improve endurance, strength, and overall well-being. All sessions were supervised by certified instructors to ensure standardization and safety (Table 2).

Attendance was high, with students participating in a median of 11 out of 12 scheduled sessions; only 6% of sessions were missed overall across the group.

In addition, participants were required to maintain a minimum of 6000 daily steps. Compliance was verified through weekly photographic submissions of pedometer records and short feedback questionnaires addressing mood, effort, and perceived challenges. This dual strategy integrated structured training with real-life activity monitoring, thereby reinforcing accountability and ecological validity.

Across all daily entries pooled over 28 days (N = 1960 observations: 70 participants × 28 days), the mean step count was 7869.39 (SD = 1445.83), showing that students, on average, exceeded the minimum daily threshold of 6000 steps. Step counts were collected through smartphone pedometer applications, including Apple Health (Apple Inc., Cupertino, CA, USA) and Google Fit (Google LLC, Mountain View, CA, USA). Participants submitted weekly screenshots of their daily step logs, and the research team transcribed each day’s totals into a centralized dataset. These daily values formed the basis for descriptive statistics and the histogram presented in Figure 1.

### 2.6. Data Analysis

The statistical analysis combined both descriptive and inferential methods to evaluate the impact of the four-week intervention program on eating behavior and emotion regulation outcomes. Descriptive statistics, including means and standard deviations (SD), were used to summarize participant characteristics and questionnaire scores at baseline and post-intervention.

To determine the distribution of the data, the Kolmogorov–Smirnov test was applied separately for each variable. Based on the outcomes of this normality assessment, appropriate statistical tests were selected. When the normality assumptions were met, paired-sample *t*-tests were conducted to evaluate the differences between pre- and post-intervention scores. For variables that deviated from normal distribution, the Wilcoxon signed-rank test was employed as a nonparametric alternative.

The internal consistency of the instruments was verified using Cronbach’s alpha. The DERS-36 total scale demonstrated an α of 0.882, indicating good reliability, while the AEBQ-35 total scale showed an α of 0.899, indicating good to excellent reliability. These values align with those reported in international validation studies, confirming the robustness of the instruments for assessing emotion regulation difficulties and eating behaviors in student populations.

For paired-sample *t*-tests, Cohen’s d was calculated to estimate effect size. For Wilcoxon signed-rank tests, effect size r was computed as Z/√N. Effect sizes were interpreted using conventional thresholds (small = 0.2, medium = 0.5, large = 0.8 for d; small = 0.1, medium = 0.3, large = 0.5 for r).

All statistical analyses were conducted using IBM SPSS Statistics, version 30.0 (IBM Corp., Armonk, NY, USA).

## 3. Results

Preliminary analyses examined the data distribution using the Kolmogorov–Smirnov test. Detailed results of the normality checks are provided in Appendix A (Table A1 and Table A2). Based on these outcomes, either parametric (paired-sample *t*-test) or nonparametric (Wilcoxon signed-rank test) methods were applied to compare pre- and post-intervention scores.

In addition to standardized scales, contextual indicators such as academic workload, psychological distress, dietary habits, cultural norms, and societal expectations were collected at baseline to provide a more detailed socio-demographic and psychosocial profile of the participants (see Table 1). These descriptive indicators were not included in the intervention analysis, as they were not reassessed post-intervention. Consequently, no changes over time are reported for these variables.

Table 3 presents the descriptive and comparative results for restrained eating, emotional eating, and external eating. No statistically significant differences were observed between pre- and post-intervention assessments across any of the three domains, indicating that the four-week intervention did not significantly alter students’ eating behavior patterns. Effect sizes indicated that all changes in eating behaviors were negligible to small (d ≤ 0.14; r ≤ 0.10), reinforcing the conclusion that the intervention did not produce meaningful short-term effects on dietary patterns.

Table 4 summarizes the results for the six dimensions of the DERS-36. Significant improvements were observed in nonacceptance of emotional responses, difficulty engaging in goal-directed behavior, impulse control difficulties, limited access to regulation strategies, and lack of emotional clarity. No significant change was detected in emotional awareness. Overall, the intervention was associated with a significant improvement in emotion regulation capacities among participants. Effect sizes confirmed that improvements in emotion regulation dimensions ranged from medium (r = 0.32) to large (r = 0.64; d = 0.70), suggesting that the intervention had a substantial positive impact on participants’ self-regulatory capacities.

## 4. Discussion

The primary objective of this study was to evaluate the effectiveness of a four-week structured intervention program in improving eating behaviors and emotion regulation among first- and second-year students of computer science and engineering at Politehnica University of Bucharest. The findings reveal that while no significant changes occurred in eating behavior dimensions, significant improvements were observed across multiple areas of emotion regulation, underscoring the value of integrating structured physical activity into academic contexts.

Eating Behavior

The results showed no statistically significant differences in Restrained Eating, Emotional Eating, or External Eating between pre- and post-intervention assessments. This aligns with previous findings that short-term interventions are often insufficient to alter ingrained eating patterns, which are shaped by long-term habits, cultural influences, and psychosocial stressors [38]. Cross-sectional studies confirm that eating behaviors among students are strongly influenced by psychological traits such as impulsivity, anxiety, and depression, with impulsivity showing the strongest predictive effect on emotional and external eating [39,40]. During stressful academic periods, such as examinations, students frequently experience deterioration in diet quality, with emotional and external eaters at higher risk of maladaptive dietary responses [9]. Taken together, these findings suggest that deeply rooted psychological traits and social pressures create resistant patterns that require longer-term or multimodal interventions—such as combining exercise with nutritional counseling and psychological support—to produce meaningful changes in eating behaviors [20,24].

Emotion Regulation

In contrast, emotion regulation indicators showed significant improvements across five of the six measured areas. Specifically, participants reported reductions in Nonacceptance of Emotional Responses, Difficulty Engaging in Goal-Directed Behavior, Impulse Control Difficulties, Limited Access to Emotion Regulation Strategies, and Lack of Emotional Clarity. These results are consistent with evidence demonstrating that structured physical activity enhances emotional resilience, self-control, and adaptive coping mechanisms [25,41].

Notably, no significant improvement was observed in the area of Lack of Emotional Awareness. This suggests that while physical activity strengthens regulatory capacity, interventions that incorporate reflective or mindfulness-based components may be required to cultivate awareness of internal states [28]. Previous research supports this interpretation. For example, Müller et al. (2021) [42] demonstrated that a single-session physical activity intervention improved mood and attention, whereas mindfulness interventions had a greater impact on executive functions. Wu et al. (2022) [43] reported that higher physical activity levels were correlated with increased use of cognitive reappraisal and decreased expressive suppression, mediated by the thickness of the rostral anterior cingulate cortex. Wisener and Khoury (2022) [44] further highlighted the role of mindfulness and self-compassion in reducing maladaptive coping. These findings collectively suggest that exercise and mindfulness may target distinct ER dimensions, which could explain the lack of change in emotional awareness in the present study.

The improvements observed can also be understood within the biopsychosocial model [45,46]. Biologically, physical activity promotes neurochemical processes linked to mood regulation [21]. Psychologically, structured routines and goal-directed activities reinforce self-efficacy and emotion regulation strategies. Socially, shared group activities enhance accountability and support, even when interventions do not explicitly focus on social outcomes [30].

Implications for Technical University Students

Computer science and engineering students are particularly vulnerable to emotional dysregulation due to academic overload, sedentary behavior, and performance-related pressures [15,23]. The present findings extend prior research on general student populations [21,47] by demonstrating that structured physical activity interventions can yield measurable psychological benefits for this subgroup. Studies suggest that technical students may respond exceptionally well to multimodal interventions: Tait et al. (2023) [48] found that both physical activity and mindfulness improved academic performance and reduced stress among engineering undergraduates, while Herbert (2022) [49] showed that yoga-based interventions immediately improved emotion processing and awareness. Cross-sectional evidence also confirms that students with higher levels of physical activity report lower levels of depression and anxiety compared to their less active peers [32]. Thus, structured activity programs may represent a scalable, practical strategy for fostering resilience within technical education contexts.

Limitations

This study has several limitations that should be acknowledged. First, although validated instruments were employed (DERS-36 and AEBQ-35), no officially validated Romanian versions currently exist. Therefore, the English versions were administered after pre-testing for comprehensibility, which may still have introduced minor biases related to language interpretation. The lack of a validated local adaptation represents a methodological limitation that should be addressed in future research [20,24].

Second, self-report measures are inherently subject to the recall effect, as participants may have misremembered or misreported their eating behaviors or emotional experiences. Moreover, step counts were obtained via self-reported smartphone screenshots, which may be prone to reporting errors or selective recording [9,32].

Third, the study design did not allow for strict control over external influences. Students may have been simultaneously exposed to other lifestyle changes, academic stressors, or informal support interventions that could have affected their outcomes [15,23]. Without a comparison group, observed improvements in emotion regulation cannot be confidently attributed solely to the intervention. Other factors, such as the academic calendar, social interactions, or seasonal variation, may also explain the results. This omission represents a significant limitation that significantly weakens causal inference [21].

Fourth, the sampling strategy relied on convenience sampling within a single faculty, which reduces the generalizability of findings to broader student populations [38]. The relatively small and homogeneous sample restricts the extent to which these results can be applied to other settings.

Fifth, while attendance was monitored, four students missed at least one intervention session. However, attrition was low, and all participants completed both pre- and post-tests. Absences may have attenuated the intervention effect size, yet the consistency of improvements across most DERS dimensions suggests a genuine program impact [41].

Sixth, differences in standard deviations (SD) between pre- and post-intervention scores (e.g., emotional eating, impulse control difficulties, limited access to emotional strategies) likely reflect a greater heterogeneity in individual responses following the program. Some students showed marked improvements, while others exhibited minimal or no change, broadening the overall variability of post-test scores [39,40]. In particular, eating habits tend to be more resistant to short-term change because they are deeply embedded in cultural norms, social contexts, and long-term routines established during adolescence and early adulthood. These cultural and social influences, already highlighted in the introduction, may help explain why the intervention failed to produce measurable improvements in dietary behaviors within a four-week timeframe [20,24,38].

Finally, this study did not employ the design of a clinical trial, which would have provided a more rigorous framework for testing the efficacy of the intervention. The absence of trial registration, randomization, and a controlled protocol further limits the strength of causal claims [45,47].

Despite these limitations, the study provides novel evidence on the potential of short-term, structured physical activity interventions to enhance emotion regulation among technical university students, laying a foundation for future, larger-scale investigations. Future studies should prioritize randomized controlled trials, larger and more diverse samples, and the development of validated instruments for the Romanian context.

Contributions and Future Directions

This study adds to the growing body of evidence supporting the integration of exercise-based interventions into higher education curricula. By demonstrating significant improvements in emotion regulation within a four-week timeframe, the findings highlight the potential of structured physical activity programs in fostering adaptive functioning among technical university students. However, the lack of change in eating behaviors emphasizes the need for longer-term and multidimensional approaches. Future research should prioritize randomized controlled designs, extend intervention duration, and integrate nutritional or psychological training components. Follow-up assessments are also essential to determine whether short-term improvements in emotion regulation translate into sustained benefits for mental health and academic performance.

## 5. Conclusions

The present study demonstrates that a four-week structured physical activity intervention produced significant improvements in several dimensions of emotion regulation among first- and second-year computer science and engineering students, while showing no measurable short-term effects on eating behaviors. Students reported better impulse control, enhanced goal-directed behavior, improved emotional clarity, and greater access to regulation strategies, reflecting more adaptive coping capacities in the face of academic stressors. These findings suggest that even a relatively brief and low-cost exercise program can enhance students’ psychological resilience, offering immediate benefits for those navigating demanding technical curricula.

From a practical standpoint, the results highlight actionable benefits for university administrators and program designers. Integrating structured, supervised exercise sessions into higher education settings could be an effective strategy to mitigate stress, enhance emotional well-being, and improve academic performance, all without requiring extensive institutional resources.

For researchers, this study offers novel evidence on the differential responsiveness of psychological versus behavioral outcomes in the context of short-term interventions. The divergence between rapid improvements in emotion regulation and the stability of eating patterns underscores the importance of tailoring interventions to specific outcome domains.

## Figures and Tables

**Figure 1 healthcare-13-02621-f001:**
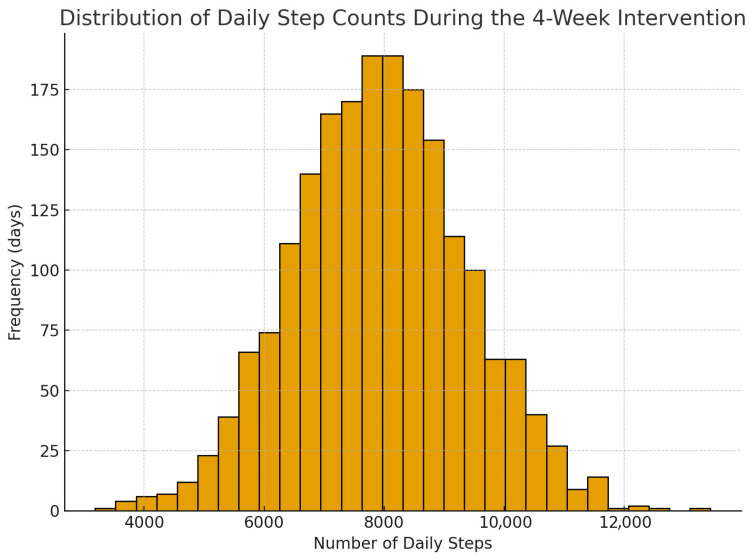
Histogram showing the distribution of daily step counts pooled across all participants and days during the 4-week intervention (N = 1960 observations). Axis labels: x-axis = “Number of daily steps”; y-axis = “Frequency (days)”. Data source: smartphone pedometer apps; daily values were transcribed from weekly screenshots submitted by participants.

**Table 1 healthcare-13-02621-t001:** Socio-Demographic and Contextual Characteristics of Participants (N = 70).

Variable	Result
Age (years)	Mean (SD): 21.03 (2.91)
Sex	Male: 40 (57%); Female: 30 (43%)
Educational Background	High School: 70 (100%)
Socioeconomic Status	Employed at Minimum Wage: 36 (51%); No Income: 34 (49%)
Living Situation	With Parents: 16 (23%); Dormitory: 32 (46%); Renting: 22 (31%)
Demographic Location	Urban: 50 (71%); Rural: 20 (29%)
Psychological Distress ^1^	Low: 26% (18); Moderate: 47% (33); High: 27% (19)
Academic Workload ^2^	Low: 5% (4); Moderate: 29% (20); High: 66% (46)
Dietary Habits ^3^	Balanced Diet: 41% (29); Irregular Eating: 34% (24); Fast Food: 25% (17)
Cultural Norms ^4^	Traditional Dietary Preferences: 61% (43); Western Influence: 39% (27)
Societal Expectations ^5^	Pressure to Maintain Body Image: 77% (54); No Pressure: 23% (16)

**Legend**. ^1^ Measured with the single-item question “How would you rate your current psychological distress?” (responses: low, moderate, high). ^2^ Measured with the single-item question “How demanding do you find your current academic workload?” (responses: low, moderate, high). ^3^ Measured with the single-item question “Which of the following best describes your dietary pattern in the last month?” (responses: balanced diet, irregular eating, predominantly fast food). ^4^ Measured with the single-item question “Which of the following dietary patterns best describes your preferences?” (responses: traditional dietary preferences, westernized influence). ^5^ Measured with the single-item question “Do you feel pressure to maintain a certain body image?” (responses: yes, no). Note: These contextual variables were collected at baseline for descriptive purposes only and were not reassessed after the intervention.

**Table 2 healthcare-13-02621-t002:** Four-Week Intervention Program Overview.

Week	Training Sessions (3×/week, 45 min)	Focus	Monitoring Component
Week 1	Aerobic circuit training + basic bodyweight exercises (push-ups, squats, planks)	Endurance and adaptation	Daily step count ≥ 6000; weekly photo submission + short feedback questionnaire
Week 2	Aerobic intervals + resistance band and light weight training	Strength, endurance, and cardiovascular improvement	Same as Week 1
Week 3	Combined aerobic + functional strength (lunges, burpees, core stability drills)	Progressive overload and functional capacity	Same as Week 1
Week 4	Mixed training (aerobic, resistance, functional) with higher intensity and group challenges	Peak performance and resilience	Same as Week 1

**Table 3 healthcare-13-02621-t003:** Pre- and Post-Intervention Scores for Eating Behavior (N = 70).

Variable	Pre (M ± SD)	Post (M ± SD)	Test	Z/t	*p*-Value	Effect Size	Result
Restrained Eating	20.81 ± 9.05	21.04 ± 8.99	Wilcoxon Signed-Rank	Z = 0.15	0.988	r = 0.02 (negligible)	n.s.
Emotional Eating	30.72 ± 12.85	28.77 ± 12.53	Wilcoxon Signed-Rank	Z = 0.80	0.425	r = 0.10 (small)	n.s.
External Eating	32.23 ± 7.19	31.50 ± 7.38	Paired-Sample *t*-Test	t = 1.17	0.246	d = 0.14 (small)	n.s.

Note: Values are presented as mean ± SD. The statistical test applied (paired-sample *t*-test or Wilcoxon signed-rank test) was determined based on the distribution of the data. Effect sizes are reported as Cohen’s d for *t*-tests and r (Z/√N) for Wilcoxon tests, interpreted using conventional thresholds (d: 0.2 small, 0.5 medium, 0.8 large; r: 0.1 small, 0.3 medium, 0.5 large). n.s. = non-significant.

**Table 4 healthcare-13-02621-t004:** Pre- and Post-Intervention Scores for Emotion Regulation (N = 70).

Variable	Pre (M ± SD)	Post (M ± SD)	Test	Z/t	*p*-Value	Effect Size	Result
Nonacceptance of Emotional Responses	13.22 ± 5.43	11.42 ± 4.84	Wilcoxon Signed-Rank	Z = 2.70	0.007	r = 0.32 (medium)	*p* < 0.01
Goal-Directed Behavior Difficulties	15.70 ± 4.08	13.45 ± 4.11	Wilcoxon Signed-Rank	Z = 4.33	< 0.001	r = 0.52 (large)	*p* < 0.01
Impulse Control Difficulties	14.75 ± 5.10	11.65 ± 3.99	Wilcoxon Signed-Rank	Z = 5.32	< 0.001	r = 0.64 (large)	*p* < 0.01
Lack of Emotional Awareness	12.95 ± 3.22	12.67 ± 2.90	Wilcoxon Signed-Rank	Z = 0.67	0.443	r = 0.08 (negligible)	n.s.
Limited Access to Regulation Strategies	18.65 ± 6.98	14.90 ± 4.87	Wilcoxon Signed-Rank	Z = 5.09	< 0.001	r = 0.61 (large)	*p* < 0.01
Lack of Emotional Clarity	12.79 ± 3.09	10.00 ± 3.17	Paired-Sample *t*-Test	t = 5.89	< 0.001	d = 0.70 (large)	*p* < 0.01

Note. Values are presented as mean ± SD. The statistical test applied (paired-sample *t*-test or Wilcoxon signed-rank test) was determined based on the distribution of the data. Effect sizes are reported as Cohen’s d for *t*-tests and r (Z/√N) for Wilcoxon tests, with thresholds for interpretation as follows: d = 0.2 small, 0.5 medium, 0.8 large; r = 0.1 small, 0.3 medium, 0.5 large. *p* < 0.01 indicates statistically significant improvement; n.s. = non-significant.

## Data Availability

The raw data supporting the conclusions of this article will be made available by the authors on request.

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
