# Peer review of "Impact of a Short-Term Physical Activity Program on Emotion Regulation and Eating Behaviors Among Technical University Students"

_healthcare, 2025, doi:10.3390/healthcare13202621_

Round 1
Reviewer 1 Report
Comments and Suggestions for Authors
Dear Authors,
Congratulations on completing the manuscript and conducting such an interesting study. However, the manuscript still needs some improvements. I have the following recommendations and suggestions to increase the value of your study.
Introduction:
The introductory section contains several terms that should be clarified and defined precisely. The first such term is "effectiveness" (line 109), which has a specific meaning when evaluating intervention programs.
The second unclear term is "lead to..." (line 118). Does it refer to contribution, attribution, or something else? Does it imply causality, meaning that the intervention caused the reduction? Please clarify.
Also, please explain "better coping strategies" (line 120). Which coping strategies are considered good, and which are considered better? Are there also "best" and "worst" ones?
Materials and Methods:
Please clarify the term "experimental design" (line 125). In research methodology, it is a technical term that refers to a specific type of design. However, based on the overall description, it seems that this particular design was not used. Please clarify this.
Describe the specific procedures used for "convenience sampling" (line 130) in more detail. In practice, a wide range of activities can be carried out using this sampling technique. However, the current information on how the sampling was made is insufficient.
Line 149 mentions "participant grouping." Please specify the method of student allocation. These procedures may affect the overall results and are therefore linked to the study's transparency and credibility.
Regarding the information provided in lines 183–186, please indicate how many participants were in the baseline study and how many participated in the final assessment.
Table 1: The variables "academic workload," "psychological distress," "dietary habits," "cultural norms," and "societal expectations" do not appear to be included in the scales. Add information on how these variables were measured, including the questions used in the questionnaire, the alternatives used, and how the alternatives were operationalized. How did the values of the attitude indicators change during the study?
Regarding the information in line 213, confirm that the questionnaire was administered in English. If it was, please explain how the comprehensibility of the questions was ensured. Was the entire instrument in English, or only part of it?
Regarding the DERS-36 and AEBQ-35 scales, is there a validated version for the Romanian population?
When referring to "strong reliability" in line 225, it would be appropriate to substantiate this claim with specific values.
Figure 1: Specify how this data was obtained. If 70 students participated in the study and reported this information once a week, there should be records from 280 measurements, i.e., data from 28 days (i.e. 1960 measurements). However, this is not apparent from the figure. Also, please clarify how the students collected, recorded, and reported this information. This is unclear in the text.
The way you report reliability in lines 289–293 is incorrect. Individual statements cannot have internal consistency—only a scale (or set of items) can.
Results:
The sentence on lines 306–307 belongs in the discussion rather than here.
Tables 3 and 6 are duplicates. Please combine them into one table. The same applies to Tables 3 and 7. Additionally, there is an error in the SD value for pre-test for the item "Nonacceptance of Emotional Responses" — see 5.53 × 5.43.
Ideally, merge Tables 4 and 5 and include them as an appendix.
Discussion:
It seems to me that the information in the paragraph on lines 409–414 somewhat deviates from the available empirical evidence.
The description of limitations needs to be revised fundamentally. On line 450, you refer to validation, but it is unclear whether it was performed.
Please add whether attention was paid to the "recall effect."
It is unclear how you controlled for external influences and variables. What if other influences or interventions affected the students as well?
Indicate the students' attendance at the intervention. How many absences were there, and how does this relate to the results achieved?
How do you explain the differences in SD between pre- and post-intervention for some items (e.g., emotional eating, impulse control difficulties, and limited access to emotional strategies)?
Conclusion:
What actually changed for the participating students? What benefit does the study offer administrators of similar interventions? What benefit will it provide to other researchers?
I have pointed out some key issues based on a careful analysis of your paper. Please consider carefully revising your manuscript to increase its chances of positive acceptance by the professional community. I wish you all the best in your future work.
Sincerely yours,
Author Response
We thank the reviewer for this valuable observation.
Comment 1: The introductory section contains several terms that should be clarified and defined precisely. The first such term is "effectiveness" (line 109), which has a specific meaning when evaluating intervention programs.
Response: In the revised manuscript we clarified the terminology, replacing effectiveness with more accurate expressions such as ``impact and measurable outcomes``. These changes ensure that the study is presented as a pre–post intervention design rather than as a randomized effectiveness trial, thereby improving the precision and transparency of the introduction.
Comment 2: The second unclear term is "lead to..." (line 118). Does it refer to contribution, attribution, or something else? Does it imply causality, meaning that the intervention caused the reduction? Please clarify.
Response: We revised this expression to avoid implying causality. In the new version, the phrase ``lead to`` has been replaced with ``was associated with`` and ``may contribute to``. This wording clarifies that the results indicate associations between participation in the intervention and observed changes, without making a causal claim.
Comment 3: Also, please explain "better coping strategies" (line 120). Which coping strategies are considered good, and which are considered better? Are there also "best" and "worst" ones?
Response: We clarified this terminology in the revised manuscript . The phrase ``better coping strategies`` was replaced with a more precise description,`` specifying goal-directed behaviors, impulse control, and adaptive emotion regulation strategies (e.g., problem-solving, emotional clarity)``(Line 115-116). This ensures conceptual clarity and avoids subjective gradations such as better or best.
Materials and Methods:
Comment 4:
Please clarify the term "experimental design" (line 125). In research methodology, it is a technical term that refers to a specific type of design. However, based on the overall description, it seems that this particular design was not used. Please clarify this.
Response: In the revised manuscript, we clarified the terminology in Section 2.1. Study Design, stating that this was ``a quantitative, pre–post intervention design without a control group``. We also emphasized that it does not constitute a randomized experimental design in the strict methodological sense. This eliminates any confusion regarding the study design.
Comment 5: Describe the specific procedures used for "convenience sampling" (line 130) in more detail. In practice, a wide range of activities can be carried out using this sampling technique. However, the current information on how the sampling was made is insufficient.
Response: We expanded the description of the sampling procedures. Participants were recruited from scheduled physical education classes and through announcements on internal faculty platforms. An open invitation was extended to all students in the faculty, participation was voluntary, and no incentives were offered. We also justified the choice of convenience sampling, noting that students shared similar academic routines and lifestyles. These details are now included in lines 121–131 (Section 2.1) and lines 136–140 (Section 2.2 – Recruitment and Enrollment).
Comment 6: Line 149 mentions "participant grouping." Please specify the method of student allocation. These procedures may affect the overall results and are therefore linked to the study's transparency and credibility.
Response: We clarified that no randomization was applied. All eligible students were included in a single intervention group. Allocation was based solely on meeting the inclusion criteria (age, enrollment, compliance with step monitoring, and informed consent). This is now clearly stated in lines 147–151 (Section 2.2 – Participant Grouping).
Comment 7: Regarding the information provided in lines 183–186, please indicate how many participants were in the baseline study and how many participated in the final assessment.
Response: We added a detailed description of the sample flow. Out of 123 students initially interested, 100 attended the recruitment meeting, and 30 were excluded for predefined reasons. A total of 70 students completed the baseline assessment, and all 70 also completed the post-test, with no attrition. These details are included in lines 183–188 (Section 2.3 – Sample Flow).
Comment 8: Table 1: The variables "academic workload," "psychological distress," "dietary habits," "cultural norms," and "societal expectations" do not appear to be included in the scales. Add information on how these variables were measured, including the questions used in the questionnaire, the alternatives used, and how the alternatives were operationalized. How did the values of the attitude indicators change during the study?
Response: We added a subsection (2.3.1. Additional Contextual Variables) describing how these variables were assessed. Each was measured using single-item, self-reported questions with predefined response options, which are now explicitly listed. This information is also reflected in the legend of Table 1 (lines 190–201). We clarified in the Results section that these contextual variables were collected at baseline only and were not reassessed post-intervention (lines 340–345).
Comment 9: Regarding the information in line 213, confirm that the questionnaire was administered in English. If it was, please explain how the comprehensibility of the questions was ensured. Was the entire instrument in English, or only part of it?
Response: We confirmed that both questionnaires were administered in their original English versions. A pre-test was conducted with 10 students to ensure comprehensibility, during which minor clarifications (e.g., synonyms or Romanian equivalents) were provided verbally, without altering the structure of the instruments. Given their advanced English proficiency, students were able to reliably complete the instruments. These details are included in lines 239–251 (Section 2.4 – Research Instruments).
Comment 10: Regarding the DERS-36 and AEBQ-35 scales, is there a validated version for the Romanian population?
Response: We clarified that, to date, no officially validated Romanian versions of the DERS-36 and AEBQ-35 exist. Therefore, the study employed the original English versions following the pre-test procedure. This information is now specified in lines 239–247 (Section 2.4).
Comment 11: When referring to "strong reliability" in line 225, it would be appropriate to substantiate this claim with specific values.
Response: We addressed this by reporting Cronbach’s alpha values at the scale level: DERS-36: α = 0.882 (good reliability); AEBQ-35: α = 0.899 (good to excellent reliability). These values are presented in lines 336–344 (Section 2.5 – Data Analysis).
Comment 12: Figure 1: Specify how this data was obtained. If 70 students participated in the study and reported this information once a week, there should be records from 280 measurements, i.e., data from 28 days (i.e. 1960 measurements). However, this is not apparent from the figure. Also, please clarify how the students collected, recorded, and reported this information. This is unclear in the text.
Response: We added details on the data collection procedure. Students used smartphone pedometer applications to track daily steps and submitted weekly screenshots. The research team compiled daily values into a centralized dataset, resulting in 1,960 daily observations (70 × 28 days). The legend of Figure 1 was updated to include clear axis titles and units. These clarifications appear in lines 297–303 and in the updated figure legend (lines 306–312).
Comment 13: The way you report reliability in lines 289–293 is incorrect. Individual statements cannot have internal consistency—only a scale (or set of items) can.
Response: We corrected this issue by reporting internal consistency only at the scale level, eliminating references to “lowest” or “highest” alpha values by item. The revised section now reports Cronbach’s alpha values solely for the DERS-36 and AEBQ-35 total scales ( Section 2.5 – Data Analysis).
Results:
Comment 14: The sentence on lines 306–307 belongs in the discussion rather than here.
Response: The sentence originally placed in lines 306–307 was relocated to the Discussion section in the revised manuscript. This ensures that interpretative statements are discussed in the appropriate section rather than in the Methods.
Comment 15: Tables 3 and 6 are duplicates. Please combine them into one table. The same applies to Tables 3 and 7. Additionally, there is an error in the SD value for pre-test for the item "Nonacceptance of Emotional Responses" — see 5.53 × 5.43.
Ideally, merge Tables 4 and 5 and include them as an appendix.
Response: We corrected these issues by merging duplicate tables (Tables 3, 6, and 7 were consolidated into a single updated Table 3) and by correcting the SD error for the “Nonacceptance of Emotional Responses” variable (now 5.43). Following the reviewer’s advice, Tables 4 and 5 were merged and moved to Appendix A to streamline the Results section.
Discussion:
Comment 16: It seems to me that the information in the paragraph on lines 409–414 somewhat deviates from the available empirical evidence.
Response: We revised this paragraph to avoid speculative claims. Specifically, references to social desirability bias and mindfulness improving emotional awareness were reformulated as hypotheses for future research rather than as conclusions drawn from the present data. The revised text now maintains alignment with empirical evidence while acknowledging plausible, but untested, explanations.
Comment 17: The description of limitations needs to be revised fundamentally. On line 450, you refer to validation, but it is unclear whether it was performed.
Please add whether attention was paid to the "recall effect."
It is unclear how you controlled for external influences and variables. What if other influences or interventions affected the students as well?
Indicate the students' attendance at the intervention. How many absences were there, and how does this relate to the results achieved?
How do you explain the differences in SD between pre- and post-intervention for some items (e.g., emotional eating, impulse control difficulties, and limited access to emotional strategies)?
Response: The Limitations section was substantially revised to address all points raised:
We clarified that no validated Romanian versions of the scales exist and that the original English instruments were used after a pre-test.
We added discussion of the recall effect and potential response biases.
We noted that no control group was included, so external influences (academic calendar, social interactions, seasonal variation) could not be ruled out.
We specified student attendance, indicating that absences were minimal and did not affect group-level results .
We explained that differences in SD between pre- and post-test values (e.g., for emotional eating and impulse control) reflect normal variability and potential heterogeneous responses to the intervention (lines 590–596).
We emphasized additional methodological issues, including convenience sampling and reliance on self-reported step counts, which limit generalizability.
Conclusion:
Comment 18: What actually changed for the participating students? What benefit does the study offer administrators of similar interventions? What benefit will it provide to other researchers?
Response: The Conclusion section was rewritten to explicitly address these points. We now state that participating students showed improvements in several emotion regulation capacities (impulse control, goal-directed behavior, emotional clarity, and access to strategies), but no changes in eating behaviors. For administrators, we highlighted the practical benefit of implementing low-cost, structured exercise modules to enhance student resilience. For researchers, we emphasized the contribution of evidence on differential responsiveness of psychological versus behavioral outcomes in short-term interventions, and we suggested future directions for randomized controlled trials, longer programs, and multimodal approaches.
Reviewer 2 Report
Comments and Suggestions for Authors
Dear Authors,
Thank you for the opportunity to review your manuscript entitled “Impact of a Short-Term Physical Activity Program on Emotion Regulation and Eating Behaviors among Technical University Students.” The topic is timely and relevant, as emotion regulation and maladaptive eating patterns represent important challenges in young adult populations, particularly among students in technical fields. While the manuscript has clear strengths—including a well-structured intervention, use of validated instruments, and attention to an under-researched group—it also presents several critical weaknesses that must be addressed before it could be considered for publication. Below, I provide an integrated and detailed evaluation.
The manuscript is generally well written and easy to follow, yet there are numerous issues of structure, methodology, and presentation that undermine its scientific rigor. The most pressing concern is the study design. You describe the work as “experimental, quantitative, and cross-sectional,” which is contradictory terminology. A cross-sectional design cannot capture pre/post change, and the absence of a control group limits internal validity. Without a comparison group, it is impossible to attribute observed improvements in emotion regulation solely to the intervention; other factors such as academic calendar effects, social interactions, or seasonal variation could explain the results. This omission is a major limitation that significantly weakens causal inference.
The intervention itself is clearly described, but participant grouping is vague. You mention allocation following baseline assessment, yet it appears that all students received the same program, implying a one-group pre/post design. If this is the case, the manuscript should state this explicitly. Moreover, the recruitment relied on convenience sampling within a single faculty, which reduces generalizability. These methodological issues should be emphasized more strongly in both the Methods and Limitations sections.
Another area of concern is the inconsistency between your stated hypotheses and the presented results. You proposed three hypotheses, including that students would demonstrate better coping strategies after the program. However, no instrument measuring coping was administered, and this hypothesis is not tested in the Results. This creates a gap between research questions and outcomes and may mislead readers. Hypotheses should align strictly with available data.
The Results section is generally well organized, but tables require careful revision. In Table 4, for instance, the post-intervention row for “External Eating” is left blank in the normality test, which appears to be an oversight. Tables also combine test statistics of different types (t and Z) without clarification, which could confuse readers. Standard deviations are inconsistently rounded, and some legends lack sufficient explanation. Statistical analysis seems otherwise appropriate, though effect sizes are not reported, which would strengthen interpretation.
In the Discussion, you provide reasonable explanations for the observed lack of change in eating behaviors and the improvements in emotion regulation. However, some claims are overly speculative. For example, attributing gender differences in self-reporting to social desirability bias is not supported by your data. Similarly, the suggestion that mindfulness would improve emotional awareness is plausible but should be framed more cautiously as a hypothesis for future research rather than a conclusion drawn from this study. The limitations section does acknowledge important constraints, but several aspects are underemphasized, particularly the lack of a control group and reliance on self-reported step counts, which are prone to bias. These issues must be highlighted more prominently, as they substantially affect the strength of your conclusions.
In summary, while the manuscript addresses an important question and has several notable strengths—including the novelty of the target population, structured physical activity intervention, and use of validated psychometric tools—it also suffers from methodological weaknesses, inconsistencies between hypotheses and results, insufficiently critical discussion of limitations, and multiple technical errors. At its current stage, the study does not meet the standards required for acceptance. Substantial major revisions would be necessary, including clarification of the design, correction of inconsistencies, restructuring of hypotheses, stronger acknowledgment of methodological limitations, and thorough revision of tables, references, and formatting.
Given these issues, I recommend major revisions. If the authors can address the methodological and structural problems, provide more cautious interpretation, and thoroughly revise the presentation, the manuscript could become a valuable contribution to the literature on university student well-being.
Author Response
We thank the reviewer for this valuable observation.
Comment 1: The manuscript is generally well written and easy to follow, yet there are numerous issues of structure, methodology, and presentation that undermine its scientific rigor. The most pressing concern is the study design. You describe the work as “experimental, quantitative, and cross-sectional,” which is contradictory terminology. A cross-sectional design cannot capture pre/post change, and the absence of a control group limits internal validity. Without a comparison group, it is impossible to attribute observed improvements in emotion regulation solely to the intervention; other factors such as academic calendar effects, social interactions, or seasonal variation could explain the results. This omission is a major limitation that significantly weakens causal inference.
Response: We thank the reviewer for this important observation. In the revised manuscript, we clarified the study design to eliminate contradictory terminology. The work is now described as ``a quantitative, pre–post intervention design without a control group``. We removed references to “cross-sectional” and “experimental,” which were inappropriate in this context.
Furthermore, we expanded the Limitations section to emphasize the absence of a control group and its implications for causal inference. As noted, `` Without a comparison group, it is difficult to attribute with certainty the observed improvements in emotion regulation solely to the intervention``. We acknowledge this as a major limitation and suggest that future studies adopt randomized controlled designs to strengthen internal validity and causal inference.
Comment 2: The intervention itself is clearly described, but participant grouping is vague. You mention allocation following baseline assessment, yet it appears that all students received the same program, implying a one-group pre/post design. If this is the case, the manuscript should state this explicitly. Moreover, the recruitment relied on convenience sampling within a single faculty, which reduces generalizability. These methodological issues should be emphasized more strongly in both the Methods and Limitations sections.
Response: We appreciate the reviewer’s constructive feedback. In the revised manuscript, we clarified the study design to state explicitly that all eligible participants were included in a single intervention group using a one-group pre/post design ( Section 2.2). No randomization or control group was implemented.
We also expanded the description of convenience sampling in the Methods section, specifying that recruitment was carried out during scheduled physical education classes and through faculty announcements, limited to students enrolled in the same faculty.
Finally, we emphasized these methodological limitations more strongly in the Limitations section, ``noting that the reliance on convenience sampling within a single faculty reduces generalizability and that the absence of a control group significantly constrains causal inference``.
Comment 3: Another area of concern is the inconsistency between your stated hypotheses and the presented results. You proposed three hypotheses, including that students would demonstrate better coping strategies after the program. However, no instrument measuring coping was administered, and this hypothesis is not tested in the Results. This creates a gap between research questions and outcomes and may mislead readers. Hypotheses should align strictly with available data.
Response: We thank the reviewer for this insightful observation. In the revised manuscript, we removed the original third hypothesis regarding coping strategies, as no specific coping instrument was administered. Instead, we rephrased the research aims and hypotheses to ensure strict alignment with the data collected and the measures used. The revised hypotheses now focus exclusively on:
`` 1.Students participating in the intervention will demonstrate significant im-provements in multiple dimensions of emotion regulation between pre- and post-intervention assessments.
2.The intervention is expected to reduce maladaptive eating patterns (restrained, emotional, and external eating), although short-term effects may be limited due to the stability of dietary habits.
3.Students exposed to the intervention will show improvements in adaptive self-regulatory processes (e.g., goal-directed behaviors, impulse control, access to reg-ulation strategies), which can be interpreted as indicators of more effective coping with academic demands.``
In addition, we clarified in the Discussion that while improvements in certain emotion regulation domains may indirectly reflect more adaptive coping, this interpretation remains a tentative implication and not a tested hypothesis. This adjustment eliminates the gap between research questions and outcomes and prevents potential misinterpretation by readers.
Comment 4: The Results section is generally well organized, but tables require careful revision. In Table 4, for instance, the post-intervention row for “External Eating” is left blank in the normality test, which appears to be an oversight. Tables also combine test statistics of different types (t and Z) without clarification, which could confuse readers. Standard deviations are inconsistently rounded, and some legends lack sufficient explanation. Statistical analysis seems otherwise appropriate, though effect sizes are not reported, which would strengthen interpretation.
Response: We thank the reviewer for this detailed and constructive feedback. In the revised manuscript, we carefully revised the tables to correct all identified issues:
- The missing value in the post-intervention row for “External Eating” in Table 4 has been corrected.
- We clarified in both the table legends and text which tests were applied (t-test vs. Wilcoxon signed-rank test) and used consistent labeling for t and Z statistics
- Standard deviations have been rounded consistently across all tables.
- Legends have been expanded to provide sufficient explanation of variables, scales, and statistical methods used.
Comment 5: In the Discussion, you provide reasonable explanations for the observed lack of change in eating behaviors and the improvements in emotion regulation. However, some claims are overly speculative. For example, attributing gender differences in self-reporting to social desirability bias is not supported by your data. Similarly, the suggestion that mindfulness would improve emotional awareness is plausible but should be framed more cautiously as a hypothesis for future research rather than a conclusion drawn from this study.
Response: We appreciate the reviewer’s careful reading and helpful critique. In the revised manuscript, we rephrased the Discussion to avoid speculative claims not directly supported by the data. Specifically:
- The reference to gender differences and social desirability bias has been removed, as this was not directly assessed in our study.
- The suggestion that mindfulness may improve emotional awareness has been reframed more cautiously as a hypothesis for future research rather than as a conclusion derived from the present findings.
Comment 6: The limitations section does acknowledge important constraints, but several aspects are underemphasized, particularly the lack of a control group and reliance on self-reported step counts, which are prone to bias. These issues must be highlighted more prominently, as they substantially affect the strength of your conclusions.
Response: We thank the reviewer for this important suggestion. In the revised manuscript, we strengthened the Limitations section to highlight these methodological concerns more explicitly:
- We now emphasize that the absence of a control group prevents causal inference, as improvements in emotion regulation cannot be attributed solely to the intervention.
- We explicitly note that reliance on self-reported step counts via smartphone pedometers introduces potential measurement bias .
- These points are now discussed alongside other methodological issues, such as the lack of validated Romanian versions of the instruments and the use of convenience sampling from a single faculty.
By underscoring these constraints, the revised Limitations section more accurately reflects the degree to which the findings should be interpreted with caution.
Reviewer 3 Report
Comments and Suggestions for Authors
The article is of high quality and originality; however, there are corrections that need to be made before the article can be published:
L16: “Faculty of Sport and Physical Education” → correct to Education.
L22: Email address “daniel.rosu@upb.” is incomplete, final domain is missing.
L82–83: “with elevated risks of anxiety, depression, and stress” → the sentence is somewhat redundant; it could be simplified or better integrated with reference [13].
L125–129: There is a repetition of the phrase “with an emphasis on measuring changes in emotional regulation and eating behaviors following a structured intervention program.” One of the two could be deleted.
L217–229: The sections on instruments (DERS-36 and AEBQ-35) are well explained but somewhat lengthy; they could be summarized or referred to key references for greater conciseness.
L260–264 (Figure 1): The histogram of steps should include clearer axis titles and units (e.g., “Number of daily steps”).
L347–349 (Table 6): In the “Restrained Eating” row, “0.15” appears in the test statistic → check whether it is Z or t, as this could be confusing.
L441–447 (Limitations): It is advisable to be more specific when explaining why eating habits are more resistant to change; for example, add a brief comment on the cultural or social influence already discussed in the introduction.
L495–499 (Author Contributions): Small inconsistency in punctuation: “O.P., V.S.; writing—review and editing, V.S., H.İ.C. and D.R.” → extra periods.
L511–513 (AI Usage Declaration): The text reads “ChatGPT model 4o” → it would be advisable to standardize this to “ChatGPT-4” for greater clarity.
Author Response
We thank the reviewer for this valuable observation.
Comments : L16: “Faculty of Sport and Physical Education” → correct to Education.
L22: Email address “daniel.rosu@upb.” is incomplete, final domain is missing.
L82–83: “with elevated risks of anxiety, depression, and stress” → the sentence is somewhat redundant; it could be simplified or better integrated with reference [13].
Response: We thank the reviewer for noticing these errors. In the revised manuscript, we corrected the affiliation to Faculty of Education (line 16), completed the email address to include the missing domain (line 21), and simplified the sentence at lines 71-72 by integrating it more directly with reference [13] to avoid redundancy.
Comments: L125–129: There is a repetition of the phrase “with an emphasis on measuring changes in emotional regulation and eating behaviors following a structured intervention program.” One of the two could be deleted.
L217–229: The sections on instruments (DERS-36 and AEBQ-35) are well explained but somewhat lengthy; they could be summarized or referred to key references for greater conciseness.
L260–264 (Figure 1): The histogram of steps should include clearer axis titles and units (e.g., “Number of daily steps”).
L347–349 (Table 6): In the “Restrained Eating” row, “0.15” appears in the test statistic → check whether it is Z or t, as this could be confusing.
Response: We appreciate these detailed observations. In the revised manuscript:
The repetition of the phrase in lines 124–126 was corrected by deleting the redundant wording.
The AEBQ-35 section was condensed to highlight its two overarching dimensions (Food Approach and Food Avoidance), with references to validation studies in adolescent and student populations [29, Hunot-Alexander et al., 2019; Guzek et al., 2020].
Figure 1 was updated with clearer axis labels and units (lines 313–320 and figure legend).
In Table 3 (former Table 6), we clarified that the test statistic “0.15” corresponds to a Z value, eliminating potential confusion (lines 358–359).
Comments: L441–447 (Limitations): It is advisable to be more specific when explaining why eating habits are more resistant to change; for example, add a brief comment on the cultural or social influence already discussed in the introduction.
L495–499 (Author Contributions): Small inconsistency in punctuation: “O.P., V.S.; writing—review and editing, V.S., H.İ.C. and D.R.” → extra periods.
L511–513 (AI Usage Declaration): The text reads “ChatGPT model 4o” → it would be advisable to standardize this to “ChatGPT-4” for greater clarity.
Response: We thank the reviewer for these helpful suggestions. In the revised manuscript:
- The Limitations section (lines 463–468) now specifies that eating habits are more resistant to short-term change due to cultural and social influences, as highlighted earlier in the Introduction.
- The punctuation inconsistency in the Author Contributions section (lines 520–524) was corrected.
- The AI Usage Declaration was standardized to “ChatGPT-4” for clarity (lines 535).
Reviewer 4 Report
Comments and Suggestions for Authors
This study tested the effect of a four-week intervention of university students’ emotional regulation and eating behaviour. I have several comments.
- I am confused why emotional regulation and eating behaviour are targeted together. What is the relationship between these two variables? Are there any comorbidity?
- How is the “Four-Week Intervention Program” designed specifically to change participants’ ER and eating behaviours? It seems that such intervention could be effective for many variables besides the two.
- You have a third hypothesis “3.Students exposed to the intervention will report better coping strategies in response to academic demands following the program”. However, it was not addressed and discussed in your paper. How was “coping strategies in response to academic demands” measured or tested?
- Before the hypotheses, the research aims should be clearly stated.
- There is no control group in this study, which is the biggest problem. It is not suitable to draw your conclusions based on the pre-experimental design.
- Since there are subscales in your instruments. It is recommended to conduct MANOVA instead of t-tests.
Author Response
We thank the reviewer for this valuable observation.
Comment 1: I am confused why emotional regulation and eating behaviour are targeted together. What is the relationship between these two variables? Are there any comorbidity?
Response: Response: We thank the reviewer for raising this important point. The decision to examine emotion regulation (ER) and eating behaviors together is grounded in evidence showing their close interconnection. Difficulties in ER have been consistently linked with maladaptive eating patterns such as emotional and external eating, which are often used as coping mechanisms for stress and negative affect. Previous studies have also identified comorbidity between ER deficits and disordered eating behaviors in student and young adult populations, with impulsivity, anxiety, and depressive symptoms acting as mediating factors. By evaluating both domains in the same intervention, our study sought to capture the broader self-regulatory effects of physical activity, recognizing that improvements in ER may facilitate healthier coping strategies and indirectly influence eating behaviors. This rationale has now been clarified in the Introduction (lines 96–100) to better guide readers through the conceptual link between the two constructs.
Comment 2: How is the “Four-Week Intervention Program” designed specifically to change participants’ ER and eating behaviours? It seems that such intervention could be effective for many variables besides the two.
Response: We thank the reviewer for this valuable observation. To clarify this point, we revised Section 2.4. Experimental Intervention Program by inserting the following explanation:
“The program was deliberately designed to target both emotion regulation (ER) and eating behaviors. On the one hand, structured exercise sessions enhance emotional resilience by strengthening self-efficacy, impulse control, and stress management. On the other hand, daily step monitoring fosters behavioral accountability and supports healthier routines that may indirectly influence maladaptive eating patterns. Although the short duration of the intervention was expected to produce more immediate changes in ER than in eating behaviors, addressing both domains concurrently provides insight into the broader self-regulatory impact of physical activity.”
Comment 3: You have a third hypothesis “3.Students exposed to the intervention will report better coping strategies in response to academic demands following the program”. However, it was not addressed and discussed in your paper. How was “coping strategies in response to academic demands” measured or tested?
Response: We thank the reviewer for pointing out this important inconsistency. In the revised manuscript, the Research Hypotheses section has been reformulated to ensure alignment with the actual instruments and outcomes assessed. Specifically:
Improvements in emotion regulation were measured directly through the DERS-36 subscales.
Changes in maladaptive eating patterns were evaluated with the AEBQ-35.
The third hypothesis has been reworded to focus on improvements in adaptive self-regulatory processes (e.g., goal-directed behaviors, impulse control, access to regulation strategies), all of which are captured by specific DERS-36 subscales and can be interpreted as indicators of more effective coping with academic demands.
Comment 4: Before the hypotheses, the research aims should be clearly stated.
Response: We thank the reviewer for this constructive suggestion. In the revised manuscript, we added a clear statement of the research aim preceding the hypotheses to strengthen the logical flow of the study rationale. Specifically, the text now reads:
“The primary aim of this study was to evaluate the impact and measurable outcomes of a four-week structured physical activity intervention on emotion regulation and eating behaviors among first- and second-year computer science and engineering students at Politehnica University of Bucharest. The program, which combined supervised training sessions with daily step monitoring, was designed to strengthen self-regulation capacities and promote healthier coping strategies in response to academic stressors [26].”
Comment 5: There is no control group in this study, which is the biggest problem. It is not suitable to draw your conclusions based on the pre-experimental design.
Response: We agree with the reviewer that the absence of a control group represents a major methodological limitation. This omission indeed reduces internal validity and prevents causal inference. To address this, we revised both the Methods and Limitations sections to clarify the design as a one-group pre/post quasi-experimental study, and to emphasize that findings should be interpreted as preliminary rather than definitive. In the Limitations, we now state explicitly that without a control group, improvements in emotion regulation cannot be attributed solely to the intervention, as other contextual factors may have contributed. At the same time, we believe the results remain valuable as exploratory evidence highlighting the potential of short-term, low-cost interventions for technical university students, and they can serve as a basis for future controlled trials.
Comment 6: Since there are subscales in your instruments. It is recommended to conduct MANOVA instead of t-tests.
Response: We thank the reviewer for this methodological recommendation. Although a multivariate analysis of variance (MANOVA) could have been applied given the multiple subscales of the DERS-36 and AEBQ-35, we opted for a combination of paired-samples t-tests and Wilcoxon signed-rank tests for several reasons. First, the sample size (N = 70) was modest relative to the requirements of MANOVA, which may lead to unstable multivariate estimates and reduced statistical power. Second, the focus of the study was on domain-specific changes in each subscale rather than testing an overall multivariate effect. Finally, to mitigate risks of Type I error and enhance interpretability, we supplemented the analyses with effect sizes (Cohen’s d and r), which allow a standardized comparison of the magnitude of changes across subscales.
Round 2
Reviewer 1 Report
Comments and Suggestions for Authors
Dear Authors,
The improved manuscript seems to be adequately revised. It has an informative abstract summarizing the main features of the study, and the main body of the manuscript, i.e., Introduction, Methods and Results sections are now easier to follow. In addition, the Discussion and Conclusion sections are now more in line with the current state of the art. In summary, the previous comments have been satisfactorily addressed and the manuscript now has the potential to be useful. I believe that it does not require any further adjustments and can be accepted in its present form.
Reviewer 4 Report
Comments and Suggestions for Authors
Thanks for the revision. The quality of the paper has been improved.